# Leveraging machine learning for enhanced and interpretable risk prediction of venous thromboembolism in acute ischemic stroke care

**Youli Jiang**[1], **Ao Li**[2], **Zhihuan Li**[3], **Yanfeng Li**[1], **Rong Li**[1], **Qingshi Zhao**[1]*, **Guisu Li**[1]*

**1** Department of Neurology, People's Hospital of Longhua, Shenzhen, China **2** Clinical Nursing Teaching and Research Section, The Second Xiangya Hospital of Central South University, Changsha, China, **3** Department of Intelligent security laboratory, Shenzhen Tsinghua University Research Institute, Shenzhen, Guangdong, China

* 66327285@qq.com (Qingshi Zhao); guisu113@163.com (Guisu Li)

## Abstract

### Background

Venous thromboembolism (VTE) is a life-threatening complication commonly occurring after acute ischemic stroke (AIS), with an increased risk of mortality. Traditional risk assessment tools lack precision in predicting VTE in AIS patients due to the omission of stroke-specific factors.

### Methods

We developed a machine learning model using clinical data from patients with acute ischemic stroke (AIS) admitted between December 2021 and December 2023. Predictive models were developed using machine learning algorithms, including Gradient Boosting Machine (GBM), Random Forest (RF), and Logistic Regression (LR). Feature selection involved stepwise logistic regression and LASSO, with SHapley Additive exPlanations (SHAP) used to enhance model interpretability. Model performance was evaluated using area under the receiver operating characteristic curve (AUC), sensitivity, specificity, positive predictive value (PPV), and negative predictive value (NPV).

### Results

Among the 1,632 AIS patients analyzed, 4.17% developed VTE. The GBM model achieved the highest predictive accuracy with an AUC of 0.923, outperforming other models such as Random Forest and Logistic Regression. The model demonstrated strong sensitivity (90.83%) and specificity (93.83%) in identifying high-risk patients. SHAP analysis revealed that key predictors of VTE risk included elevated D-dimer levels, premorbid mRS, and large vessel occlusion, offering clinicians valuable insights for personalized treatment decisions.

**Data availability statement:** The dataset underlying this study contains sensitive patient information, and making it publicly available would compromise patient privacy in accordance with the protocol approved by an institutional ethics committee. However, researchers interested in accessing the data may contact Changyu Li, Ethics Committee member, via email (yeslcy888@163.com). Access to the data will be granted under specific terms that ensure the protection of patient confidentiality

**Funding:** This study was supported by the Scientific Research Projects of Medical and Health Institutions of Longhua District, Shenzhen (grant number 2024067) in the form of a grant awarded to Y. Jiang. The specific roles of this author are articulated in the 'author contributions' section. The funders had no role in study design, data collection and analysis, decision to publish, or preparation of the manuscript.

**Competing interests:** The authors declares no potential conflicts of interest with respect to the research, authorship, and publication of this article.

## Conclusion

This study provides an accurate and interpretable method to predict VTE risk in patients with AIS using the GBM model, potentially improving early detection rates and reducing morbidity. Further validation is needed to assess its broader clinical applicability.

## Introduction

Stroke remains a paramount health challenge worldwide, ranking as the second leading cause of mortality and contributing to approximately 6.1 million deaths each year. It also stands as a principal cause of long-term disability [1]. Among its subtypes, acute ischemic strokes (AIS) predominate, significantly influencing the convalescence and functional recovery of patients [2]. A critical concern in the aftermath of an ischemic stroke is the development of venous thromboembolism (VTE), a severe complication typically arising within the first two weeks post-event, with its highest risk noted in the initial seven days [3, 4]. The occurrence of VTE is closely linked with an escalated risk of mortality within three months following a stroke, highlighting its importance as a preventable aspect of post-stroke care management [5, 6].

Conventional risk assessment tools such as the Caprini scoring system and the Padua Prediction Score are commonly applied across various patient groups but show significant limitations in stroke populations [7]. Traditional tools like the Caprini and Padua scores, though widely used in general and surgical populations, often omit critical stroke-specific factors like neurological status, stroke severity (assessed by NIHSS), and interventions such as thrombolysis or thrombectomy, leading to reduced predictive accuracy [8]. Furthermore, current VTE risk models for stroke patients often fail to incorporate evolving clinical factors, reducing predictive accuracy [9–11]. As a result, these models often underestimate the true risk in stroke patients, leading to inaccurate predictions. These deficiencies hinder the models' effectiveness in practical, clinical applications. This point has already been mentioned and can be removed to avoid redundancy. You can omit this sentence entirely since it overlaps with earlier statements. In particular, they often fail to capture dynamic, patient-specific factors such as functional impairments and response to therapies. Our machine learning-based model addresses the heterogeneous nature of stroke patients by integrating both baseline and dynamic factors, offering superior risk stratification and predictive accuracy, especially compared to traditional models like logistic regression that were developed for broader patient populations [12, 13].

There exists a notable gap in the research landscape concerning the development of machine learning models that not only predict VTE with high precision but also adhere to the stringent criteria set forth by the Predictive Model Bias Risk Assessment Tool (PROBAST) [14]. Furthermore, the literature reveals a conspicuous scarcity of studies employing advanced machine learning techniques in conjunction with SHapley Additive exPlanations (SHAP) interpretability algorithms to forecast VTE subsequent to AIS [15]. This study aims to bridge this gap through rigorous data collection and preprocessing efforts, alongside the application of sophisticated machine learning methodologies. Our model is specifically designed to enhance the predictive accuracy for DVT in AIS patients, incorporating a broad spectrum of stroke-specific variables and advanced treatment interventions. This innovative approach provides clinicians with a precise, interpretable framework to identify high-risk patients and formulate personalized interventions, ultimately reducing the incidence of lower extremity DVT and improving post-stroke outcomes.

## Materials and methods

### Study Population

The dataset for this investigation was derived from the Shenzhen Neurological Disease System Platform, an extensive repository that has been methodically aggregating detailed data on ischemic stroke patients since 2021, encompassing sociodemographic characteristics, precise details regarding pre-hospital onset, in-hospital diagnostic findings, treatment records, and laboratory test outcomes from 20 affiliated hospitals. Patients were eligible for inclusion if they were aged 18 years or older, admitted between December 2021 and December 2023, diagnosed with acute ischemic stroke confirmed using magnetic resonance imaging (MRI) or computed tomography angiography (CTA), admitted within seven days of symptom onset, and met the International Classification of Diseases, Tenth Revision (ICD-10) criteria. Exclusions were made for patients with transient ischemic attacks, subarachnoid hemorrhage, brain tumors, cerebral venous thrombosis, those diagnosed with distal deep vein thrombosis (DVT) prior to admission, or with a history of pulmonary embolism; additional exclusion criteria included severe renal or hepatic dysfunction, active cancer or undergoing chemotherapy/radiation therapy, known hypercoagulable states, prior use of anticoagulant therapy, incomplete or missing essential medical records, and loss to follow-up during hospitalization. The dataset was split into a training set (December 2021 to June 2023) and a test set (July 2023 to December 2023). To ensure robust hyperparameter tuning, we employed 10-fold cross-validation within the training set. During cross-validation, the training set was further divided into 90% training folds and 10% validation folds in each iteration. This process ensured that hyperparameter tuning was based on unseen data. The test set remained completely independent and was used only for the final evaluation of model performance. The platform's data management ensured data anonymization and quality control. Data for this research were accessed on January 24, 2024, from diagnostic and treatment documentation recorded by neurology specialists. The final cohort included 1,632 patients admitted between December 2021 and December 2023. The median age was 58 years (IQR 49–68), with 72.1% male and 96.9% Han Chinese patients. Common comorbidities were diabetes mellitus (21.2%), atrial fibrillation (4.0%), and prior stroke (18.2%)

### Predictor variables and outcomes

This study aimed to construct a predictive model for VTE occurrence following AIS using a comprehensive dataset representing diverse AIS patient profiles. Patients enrolled from December 2021 to December 2023 in the Shenzhen Neurological Disease System Platform were analyzed. The selection of potential predictive variables was confined to characteristics documented within the initial three days of hospital admission. The predictive variables included demographic data such as gender, age, and ethnicity; clinical factors like diabetes mellitus, atrial fibrillation, and a history of cerebral infarction; treatment-related factors including endovascular treatment, thrombolytic therapy, and prehospital medications; and laboratory results such as D-dimer levels, fibrinogen, and international normalized ratio (INR). A complete list of variables is detailed in Table 1 and Supplementary S1 Table.

VTE diagnoses were confirmed using color Doppler ultrasound and pulmonary CT angiography (PCTA), with imaging results cross-referenced against clinical assessments. A stratified screening methodology was employed for stroke patients manifesting potential symptoms of VTE during their hospital stay, including but not limited to leg pain or swelling, localized warmth, dyspnea, or chest discomfort. This screening approach identified high-risk individuals for further evaluation with color Doppler ultrasound or PCTA.

**Table 1. Demographic and Baseline Characteristics by VTE Status.**

| Variable | Category | VTE (n = 68; %): Median [IQR 25%-75%] | Non-VTE (n = 1564; %): Median [IQR 25%-75%] | P-Value |
|---|---|---|---|---|
| Gender | Female | 30 (6.58) | 426 (93.42) | 0.004 |
| | Male | 38 (3.23) | 1138 (96.77) | |
| Ethnicity | Han | 63 (3.98) | 1519 (96.02) | 0.082 |
| | Others | 5 (10.00) | 45 (90.00) | |
| Age | | 69.00 (60.00-79.00) | 58.00 (49.00-67.00) | <0.001 |
| Height (cm) | | 163.60 (156.78-170.00) | 168.00 (159.00-170.00) | 0.041 |
| Weight (kg) | | 67.00 (60.00-72.00) | 65.00 (54.83-73.54) | 0.468 |
| Smoking | No | 56 (5.09) | 1044 (94.91) | 0.011 |
| | Yes | 12 (2.26) | 520 (97.74) | |
| Drinking | No | 63 (5.45) | 1093 (94.55) | <0.001 |
| | Yes | 5 (1.05) | 471 (98.95) | |
| DM | No | 62 (4.82) | 1224 (95.18) | 0.016 |
| | Yes | 6 (1.73) | 340 (98.27) | |
| Hyperlipidemia | No | 66 (4.94) | 1271 (95.06) | 0.002 |
| | Yes | 2 (0.68) | 293 (99.32) | |
| Atrial fibrillation | No | 61 (3.89) | 1506 (96.11) | 0.016 |
| | Yes | 7 (10.77) | 58 (89.23) | 0.016 |
| History of cerebral infarction | No | 55 (4.12) | 1280 (95.88) | 0.968 |
| | Yes | 13 (4.38) | 284 (95.62) | |
| Anemia | No | 57 (3.52) | 1564 (96.48) | <0.001 |
| | Yes | 10 (90.90) | 1 (9.09) | |
| Other comorbid conditions | No | 64 (5.38) | 1125 (94.62) | <0.001 |
| | Yes | 4 (0.90) | 439 (99.10) | |
| ECG results | Normal | 48 (3.14) | 1482 (96.86) | <0.001 |
| | Abnormal | 20 (19.61) | 82 (80.39) | |
| Prehospital medication | No | 55 (7.70) | 659 (92.30) | <0.001 |
| | Yes | 13 (1.42) | 905 (98.58) | |
| Premorbid mRS | 0-1 | 17 (1.14) | 1476 (98.86) | <0.001 |
| | 2-3 | 19 (20.43) | 74 (79.57) | |
| | 4-5 | 32 (69.57) | 14 (30.43) | |
| mRS after admission | 0-1 | 1 (0.21) | 481 (99.79) | <0.001 |
| | 2-3 | 22 (3.42) | 621 (96.58) | |
| | 4-5 | 45 (8.88) | 462 (91.12) | |
| NIHSS onset | 0-4 | 6 (0.60) | 998 (99.40) | <0.001 |
| | 5-14 | 51 (9.55) | 483 (90.45) | |
| | 15-20 | 7 (11.86) | 52 (88.14) | |
| | 21-42 | 4 (11.43) | 31 (88.57) | |
| NIHSS after admission | 0-4 | 6 (0.60) | 1000 (99.40) | <0.001 |
| | 5-14 | 47 (9.18) | 465 (90.82) | |
| | 15-20 | 11 (13.92) | 68 (86.08) | |
| | 21-42 | 4 (11.43) | 31 (88.57) | |
| GCS | 13-15 | 14 (25.45) | 41 (74.55) | <0.001 |
| | 9-12 | 2 (2.47) | 79 (97.53) | |
| | 3-8 | 52 (3.48) | 1444 (96.52) | |
| SBP | | 83.50 (75.25-100.75) | 88.00 (78.00-100.00) | 0.164 |
| DBP | | 142.50 (121.00-157.75) | 149.00 (133.00-168.00) | |

*(Continued)*

**Table 1.** (Continued)

| Variable | Category | VTE (n = 68; %): Median [IQR 25%-75%] | Non-VTE (n = 1564; %): Median [IQR 25%-75%] | P-Value |
|---|---|---|---|---|
| TOAST classification | Large vessel occlusion | 5 (1.22) | 404 (98.78) | 0.001 |
| | Small vessel occlusive stroke | 12 (2.17) | 540 (97.83) | 0.006 |
| | Cardioembolic stroke | 4 (0.85) | 469 (99.15) | <0.001 |
| | Other causes of stroke | 16 (11.11) | 128 (88.89) | <0.001 |
| | Unexplained stroke | 31 (57.41) | 23 (42.59) | <0.001 |
| Weakness | No | 41 (7.01) | 544 (92.99) | <0.001 |
| | Yes | 27 (2.58) | 1020 (97.42) | |
| Dysarthria | No | 57 (5.50) | 980 (94.50) | 0.001 |
| | Yes | 11 (1.85) | 584 (98.15) | |
| Other symptoms | No | 67 (5.65) | 1118 (94.35) | <0.001 |
| | Yes | 1 (0.22) | 446 (99.78) | |
| Dizziness | No | 64 (4.76) | 1281 (95.24) | <0.001 |
| | Yes | 4 (1.39) | 283 (98.61) | |
| Paresthesia | No | 68 (4.52) | 1438 (95.48) | 0.027 |
| | Yes | 0 (0.00) | 126 (100.00) | |
| Headache | No | 64 (3.98) | 1546 (96.02) | 0.006 |
| | Yes | 4 (18.18) | 18 (81.82) | |
| Dizzy | No | 65 (4.85) | 1274 (95.15) | 0.005 |
| | Yes | 3 (1.02) | 290 (98.98) | |
| Convulsion | No | 65 (4.01) | 1556 (95.99) | 0.002 |
| | Yes | 3 (27.27) | 8 (72.73) | |
| Consciousness status | No | 49 (3.16) | 1501 (96.84) | <0.001 |
| | Yes | 19 (23.17) | 63 (76.83) | |
| Other symptoms | No | 67 (5.65) | 1118 (94.35) | <0.001 |
| | Yes | 1 (0.22) | 446 (99.78) | |
| Symptomatic treatment | No | 23 (6.52) | 330 (93.48) | 0.019 |
| | Yes | 45 (3.52) | 1234 (96.48) | |
| EVT | No | 51 (3.42) | 1440 (96.58) | <0.001 |
| | Yes | 17 (12.06) | 124 (87.94) | |
| Thrombolytic therapy | No | 55 (3.98) | 1327 (96.02) | 0.474 |
| | Yes | 13 (5.20) | 237 (94.80) | |
| Lymphocyte count | | 1.30 (0.88-2.02) | 1.79 (1.34-2.33) | 0.033 |
| hsCRP | | 18.93 (5.44-38.18) | 2.70 (1.32-6.60) | <0.001 |
| INR | | 1.06 (0.98-1.16) | 1.00 (0.96-1.05) | 0.005 |
| Fibrinogen | | 3.06 (2.63-3.59) | 3.41(2.91-4.02) | <0.001 |
| D-dimer | | 5.42 (1.34-18.66) | 0.33 (0.19-0.67) | <0.001 |
| Alanine aminotransferase | | 20.00 (14.10-29.00) | 23.00 (16.00-35.75) | 0.045 |
| LDLC | | 2.69 (2.07-3.29) | 3.02(2.46-3.55) | 0.001 |
| Aspirin | No | 39 (16.05) | 204 (83.95) | <0.001 |
| | Yes | 29 (2.09) | 1360 (97.91) | |
| Clopidogrel | No | 34 (14.17) | 206 (85.83) | <0.001 |
| | Yes | 34 (2.44) | 1358 (97.56) | |
| Heparin | No | 30 (1.89) | 1559 (98.11) | <0.001 |
| | Yes | 38 (88.37) | 5 (11.63) | |
| Enoxaparin | No | 44 (2.74) | 1563 (97.26) | <0.001 |
| | Yes | 24 (96.00) | 1 (4.00) | |

*(Continued)*

**Table 1.** (Continued)

| Variable | Category | VTE (n = 68; %): Median [IQR 25%-75%] | Non-VTE (n = 1564; %): Median [IQR 25%-75%] | P-Value |
|---|---|---|---|---|
| Low molecular weight heparin | No | 57 (3.52) | 1561 (96.48) | <0.001 |
| | Yes | 11 (78.57) | 3 (21.43) | |
| Unfractioted heparin | No | 67 (4.11) | 1563 (95.89) | 0.14 |
| | Yes | 1 (50.00) | 1 (50.00) | |
| Warfarin | No | 65 (4.04) | 1543 (95.96) | 0.123 |
| | Yes | 3 (12.50) | 21 (87.50) | |
| Rivaroxaban | No | 43 (2.86) | 1462 (97.14) | <0.001 |
| | Yes | 25 (19.69) | 102 (80.31) | |
| Sulfonylureas | No | 67 (4.74) | 1347 (95.26) | 0.006 |
| | Yes | 1 (0.46) | 217 (99.54) | |
| Glycosidase inhibitor | No | 68 (4.78) | 1355 (95.22) | 0.002 |
| | Yes | 0 (0.00) | 209 (100.00) | |
| Anti-infective treatment | No | 38 (2.55) | 1451 (97.45) | <0.001 |
| | Yes | 30 (20.98) | 113 (79.02) | |
| Lipid medicine | No | 14 (23.73) | 45 (76.27) | <0.001 |
| | Yes | 54 (3.43) | 1519 (96.57) | |
| Anti-platelet therapy during hospitalization | Yes | 42 (2.73) | 1496 (97.27) | <0.001 |
| | No | 26 (27.66) | 68 (72.34) | |
| Anticoagulant therapy during hospitalization | Yes | 18 (1.25) | 1425 (98.75) | <0.001 |
| | No | 50 (26.46) | 139 (73.54) | |
| Antilipidemic drugs during hospitalization | No | 14 (25.93) | 40 (74.07) | <0.001 |
| | Yes | 54 (3.42) | 1524 (96.58) | |
| Antidiabetic treatment during hospitalization | No | 60 (5.07) | 1123 (94.93) | 0.005 |
| | Yes | 8 (1.78) | 441 (98.22) | |
| Chinese medicines during hospitalization | No | 39 (2.46) | 1546 (97.54) | <0.001 |
| | Yes | 29 (61.70) | 18 (38.30) | |
| Intracranial artery stenosis | No | 62 (4.86) | 1215 (95.14) | 0.013 |
| | Yes | 6 (1.69) | 349 (98.31) | |

## Data processing and feature selection

In this study, we harnessed advanced machine learning techniques, notably the K-nearest neighbor (KNN) algorithm and the synthetic minority oversampling technique (SMOTE), to refine our dataset, thereby augmenting the predictive accuracy for VTE risk [16, 17]. We first split the dataset into training and test sets; SMOTE was applied only to the training set to prevent data leakage and reduce the risk of overfitting. The KNN algorithm was used to impute missing values separately on the training and test sets, preserving the dataset's integrity. Regarding the missing number, 30% of the variables were eliminated in this study and were not included in the data analysis (S1 Table). By generating synthetic samples for minority classes, SMOTE amplified the representation of these underrepresented groups within the dataset, ensuring a more equitable representation of all classes and improving sensitivity and accuracy in predicting VTE occurrences, a relatively rare but clinically significant event. By employing these data processing and feature selection techniques, we established a robust foundation for developing a highly accurate and generalizable VTE risk prediction model.

## Model development and performance evaluation

To develop an accurate predictive model for VTE risk following AIS, we integrated statistical methods with domain knowledge in the model development process. Hyperparameter tuning was conducted using grid search within a 10-fold cross-validation framework applied to the training set, ensuring robust evaluation and minimizing overfitting.During each fold, the training set was split into 90% for model training and 10% as a validation fold to evaluate hyperparameter configurations, ensuring the models were optimized using unseen data during tuning. Detailed parameter ranges for each model are provided in Supplementary S3 File. An independent test set, consisting of unseen data, was reserved exclusively for the final evaluation to ensure generalizability. A range of machine learning algorithms was explored, including Logistic Regression (LR), Naive Bayes (NB), Decision Trees (DT), Random Forest (RF), GBM, Extreme Gradient Boosting (XGB), and Support Vector Machines (SVM). The evaluation of model performance considered multiple metrics, including AUC, sensitivity, specificity, positive predictive value (PPV), negative predictive value (NPV), calibration plots to assess agreement between predicted probabilities and observed outcomes, and the Precision-Recall (PR) curve to further evaluate performance in imbalanced datasets. This comprehensive approach ensured a reliable and interpretable model selection process.

## Statistical analysis

We conducted descriptive statistical analysis using Chi-square tests for categorical variables, T-tests for normally distributed continuous variables, and Mann-Whitney U tests for non-normal variables. This phase aimed to isolate independent predictors of distal DVT in patients experiencing acute stroke, with variables demonstrating P-values < 0.05 advancing to the subsequent feature selection stage. Feature selection was conducted using stepwise forward logistic regression and LASSO to identify relevant predictors, integrating both statistical significance and clinical relevance [18]. Visualization techniques, including scatter plots and ROC curve analysis, were used to illustrate algorithm efficacy. Additionally, SHAP algorithm analysis was employed to enhance the interpretability of model features, particularly for the optimal model. Data analysis was performed in Python.

## Ethics approval and consent to participate

This study was conducted in accordance with the Declaration of Helsinki and was granted an exemption from ethics approval by the Ethics Review Committee of Shenzhen Longhua District People's Hospital. The committee determined that the research involved minimal risk to participants and utilized anonymized data collected as part of routine clinical care. Therefore, individual informed consent was waived. Confidentiality and data privacy were strictly maintained; all personal identifiers were removed, and data were anonymized prior to analysis. Access to the data was restricted to authorized research personnel, and all data were securely stored in password-protected databases, complying with relevant data protection regulations.

# Results

## Characteristics of study population

The cohort comprised 1,632 subjects, among which the incidence of VTE was 4.17% (n = 68), with a notable predominance of female patients. The median age of individuals diagnosed with VTE was 69.00 years, significantly older than their non-VTE counterparts, who had a median age of 58.00 years (p < 0.001). Detailed demographic and clinical characteristics of the study population are delineated in Table 1.

For analytical purposes, the participants were stratified into a training set (n = 1,142) and a test set (n = 490). The composition was predominantly female (71.98% in the training set, 72.24% in the test set) and of Han ethnicity (97.20% in the training set, 96.33% in the test set), with other demographic and clinical attributes showing no significant differences between the two groups, thus ensuring a balanced representation of stroke-related outcomes (Table 2).

## Correlation of variables with clinical outcome

Univariate analysis revealed significant associations between VTE occurrence and several variables, including Gender (p = 0.004), Age (p < 0.001), Height (p = 0.041), Smoking status (p = 0.011), Alcohol consumption (p < 0.001), DM (p = 0.016), Hyperlipidemia (p = 0.002), Atrial fibrillation (p = 0.016), Anemia (p < 0.001), Dysarthria (p = 0.001), ECG findings (p < 0.001), prehospital medication (p < 0.001), pre-morbid Modified Rankin Scale (mRS) (p < 0.001), mRS after admission (p < 0.001), NIHSS at onset (p < 0.001), NIHSS after admission (p < 0.001), GCS (p < 0.001), TOAST classification, weakness, consciousness status, Endovascular treatment (EVT), D-dimer, and LDLC, among other variables. These detailed comparison results are provided in Table 1.

The LASSO model, employing a cross-validation mechanism, fine-tuned the regularization strength (alpha) over a logarithmic scale from $10^{-6}$ to $10^{1}$, facilitating precise feature selection. A specified random_state parameter ensured the reproducibility of the findings. The model's comprehensive analysis underscored the significance of variables such as Pre-morbid mRS, unexplained stroke, in-hospital medications, among others, affirming their relevance to the study's aims (Fig 1E and F). Concurrently, stepwise forward logistic regression was utilized to identify pertinent variables for univariate analysis, complementing the LASSO model's feature selection to guarantee a comprehensive set of predictors for model development. The outcomes of this meticulous variable screening process are presented in S2 Table.

## Development and validation of predictive models

Our study employed t-distributed Stochastic Neighbor Embedding (t-SNE) for dimensionality reduction, facilitating a detailed visualization of the distribution patterns between VTE and non-VTE cases within our training dataset (Fig 1). The initial dataset displayed a mixed distribution (Fig 1A), which became sparser following random undersampling (Fig 1B). Conversely, distributions post-oversampling and the application of SMOTE-NC illustrated a more dispersed pattern (Fig 1C and 1D), indicating the significant impact of sampling techniques on data representation.

Through ten-fold cross-validation, we meticulously evaluated the performance of seven distinct machine learning models. The results showcased the GBM leading with an AUC score of 0.974, followed by RF at 0.925, DT at 0.883, XGB at 0.879, NB at 0.858, LR at 0.854, and SVM at 0.853 (Fig 2A). In terms of performance metrics beyond AUC, the models also demonstrated varying levels of sensitivity and specificity, with GBM achieving the highest sensitivity (0.923) and specificity (0.938), affirming its superior predictive power. In addition to AUC, we evaluated the performance of the GBM model using the PR curve, given the class imbalance in our dataset. The PR curve demonstrated an average precision (AP) score of 0.925, underscoring the model's capability to effectively identify the minority class. Detailed PR curve analysis is provided in Supplementary S4 File. These results underscore the importance of hyperparameter tuning, which was conducted through grid search optimization within the cross-validation framework. The optimal hyperparameters for each model are presented in S3 File, providing detailed insights into the model configurations that resulted in the highest performance.

**Table 2. Distribution of Demographic and Clinical Variables in Training and Test Sets.**

| Variable | Category | Training Set (n = 1142): %, Median [IQR 25%-75%] | Test Set (n = 490): %, Median [IQR 25%-75%] | P-Value |
|---|---|---|---|---|
| Gender | Male | 822 (71.98) | 354 (72.24) | 0.960 |
| | Female | 320 (28.02) | 136 (27.76) | 0.960 |
| Ethnicity | Han | 1110 (97.20) | 472 (96.33) | 0.436 |
| | Others | 32 (2.80) | 18 (3.67) | 0.436 |
| Age | | 58.0 (49.0-68.0) | 59.0 (51.0-67.0) | 0.286 |
| Height | | 168.0 (159.0-170.1) | 168.0 (160.0-170.3) | 0.148 |
| Weight | | 66.93 (59.0-72.0) | 67.32 (60.0-71.922) | 0.293 |
| Smoking | No | 764 (66.90) | 336 (68.57) | 0.547 |
| | Yes | 378 (33.10) | 154 (31.43) | 0.547 |
| Drinking | No | 806 (70.58) | 350 (71.43) | 0.774 |
| | Yes | 336 (29.42) | 140 (28.57) | 0.774 |
| DM | No | 904 (79.16) | 382 (77.96) | 0.633 |
| | Yes | 238 (20.84) | 108 (22.04) | 0.633 |
| Hyperlipidemia | No | 941 (82.40) | 396 (80.82) | 0.489 |
| | Yes | 201 (17.60) | 94 (19.18) | 0.489 |
| Atrial fibrillation | No | 1101 (96.41) | 466 (95.10) | 0.271 |
| | Yes | 41 (3.59) | 24 (4.90) | 0.271 |
| History of cerebral infarction | No | 941 (82.40) | 394 (80.41) | 0.376 |
| | Yes | 201 (17.60) | 96 (19.59) | 0.376 |
| Anemia | No | 1134 (99.30) | 487 (99.39) | 1.000 |
| | Yes | 8 (0.70) | 3 (0.61) | 1.000 |
| Hyperlipidemia | No | 941 (82.40) | 396 (80.82) | 0.489 |
| | Yes | 201 (17.60) | 94 (19.18) | 0.489 |
| Other comorbid conditions | No | 819 (71.72) | 370 (75.51) | 0.129 |
| | Yes | 323 (28.28) | 120 (24.49) | 0.129 |
| ECG | Normal | 1070 (93.70) | 460 (93.88) | 0.978 |
| | Abnormal | 72 (6.30) | 30 (6.12) | 0.978 |
| Prehospital medication | No | 648 (56.74) | 270 (55.10) | 0.577 |
| | Yes | 494 (43.26) | 220 (44.90) | 0.577 |
| Pre-morbid mRS | 0-1 | 1048 (91.77) | 445 (90.82) | 0.129 |
| | 2-3 | 58 (5.08) | 35 (7.14) | 0.129 |
| | 4-6 | 36 (3.15) | 10 (2.04) | 0.129 |
| mRS after admission | 0-1 | 440 (38.53) | 203 (41.43) | 0.540 |
| | 2-3 | 361 (31.61) | 146 (29.80) | 0.540 |
| | 4-6 | 341 (29.86) | 141 (28.78) | 0.540 |
| NIHSS onset | 0-4 | 699 (61.21) | 305 (62.24) | 0.409 |
| | 5-14 | 372 (32.57) | 162 (33.06) | 0.409 |
| | 15-20 | 42 (3.68) | 17 (3.47) | 0.409 |
| | 21-42 | 29 (2.54) | 6 (1.22) | 0.409 |
| NIHSS during hospitalization | 0-4 | 699 (61.21) | 307 (62.65) | 0.282 |
| | 5-14 | 355 (31.09) | 157 (32.04) | 0.282 |
| | 15-20 | 59 (5.17) | 20 (4.08) | 0.282 |
| | 21-42 | 29 (2.54) | 6 (1.22) | 0.282 |
| GCS | 13-15 | 1042 (91.24) | 454 (92.65) | 0.398 |
| | 9-12 | 57 (4.99) | 24 (4.90) | 0.398 |
| | 3-8 | 43 (3.77) | 12 (2.45) | 0.398 |

*(Continued)*

**Table 2.** (Continued)

| Variable | Category | Training Set (n = 1142): %, Median [IQR 25%-75%] | Test Set (n = 490): %, Median [IQR 25%-75%] | P-Value |
|---|---|---|---|---|
| SBP | | 88.0 (78.0-100.0) | 87.0 (78.0-99.0) | 0.129 |
| DBP | | | | |
| TOAST classification | Large vessel occlusion | 288 (25.22) | 121 (24.69) | 0.871 |
| | Small vessel occlusive stroke | 386 (33.80) | 166 (33.88) | 1.000 |
| | Cardioembolic stroke | 330 (28.90) | 143 (29.18) | 0.954 |
| | Other causes of stroke | 96 (8.41) | 48 (9.80) | 0.417 |
| | Unexplained stroke | 42 (3.68) | 12 (2.45) | 0.262 |
| Weakness | No | 727 (63.66) | 320 (65.31) | 0.562 |
| | Yes | 415 (36.34) | 170 (34.69) | 0.562 |
| Dysarthria | No | 748 (65.50) | 289 (58.98) | 0.014 |
| | Yes | 394 (34.50) | 201 (41.02) | 0.014 |
| Other symptoms | No | 820 (71.80) | 365 (74.49) | 0.292 |
| | Yes | 322 (28.20) | 125 (25.51) | 0.292 |
| Dizziness | No | 944 (82.66) | 405 (82.65) | 1.000 |
| | Yes | 198 (17.34) | 85 (17.35) | 1.000 |
| Paresthesia | No | 1053 (92.21) | 453 (92.45) | 0.947 |
| | Yes | 89 (7.79) | 37 (7.55) | 0.947 |
| Headache | No | 1126 (98.60) | 484 (98.78) | 0.961 |
| | Yes | 16 (1.40) | 6 (1.22) | 0.961 |
| Dizzy | No | 936 (81.96) | 403 (82.24) | 0.947 |
| | Yes | 206 (18.04) | 87 (17.76) | 0.947 |
| Convulsion | No | 1135 (99.39) | 486 (99.18) | 0.896 |
| | Yes | 7 (0.61) | 4 (0.82) | 0.896 |
| Other symptoms | No | 820 (71.80) | 365 (74.49) | 0.292 |
| | Yes | 322 (28.20) | 125 (25.51) | 0.292 |
| Symptomatic treatment | No | 900 (78.81) | 379 (77.35) | 0.554 |
| | Yes | 242 (21.19) | 111 (22.65) | 0.554 |
| EVT | No | 1040 (91.07) | 451 (92.04) | 0.586 |
| | Yes | 102 (8.93) | 39 (7.96) | 0.586 |
| Thrombolytic therapy | No | 970 (84.94) | 412 (84.08) | 0.715 |
| | Yes | 172 (15.06) | 78 (15.92) | 0.715 |
| Lymphocyte count | | 1.77 (1.32-2.33) | 1.78 (1.34-2.297) | 0.971 |
| hsCRP | | 2.86 (1.40-7.69) | 2.545 (1.31-6.38) | 0.033 |
| INR | | 1.01 (0.95-1.06) | 1.00 (0.96-1.06) | 0.507 |
| fibrinogen | | 3.09 (2.65-3.64) | 3.0385 (2.63-3.52) | 0.189 |
| D-dimer | | 0.34 (0.19-0.77) | 0.352 (0.21-0.71) | 0.886 |
| alanine aminotransferase | | 20.0 (14.73-30.0) | 20.0 (15.0-29.0) | 0.754 |
| LDLC | | 3.0 (2.43-3.55) | 3.01 (2.47-3.52) | 0.860 |
| Aspirin | No | 964 (84.41) | 425 (86.73) | 0.258 |
| | Yes | 178 (15.59) | 65 (13.27) | 0.258 |
| Clopidogrel | No | 968 (84.76) | 424 (86.53) | 0.397 |
| | Yes | 174 (15.24) | 66 (13.47) | 0.397 |
| Heparin | No | 1111 (97.29) | 478 (97.55) | 0.890 |
| | Yes | 31 (2.71) | 12 (2.45) | 0.890 |
| Enoxaparin | No | 1122 (98.25) | 485 (98.98) | 0.378 |
| | Yes | 20 (1.75) | 5 (1.02) | 0.378 |

*(Continued)*

**Table 2.** (Continued)

| Variable | Category | Training Set (n = 1142): %, Median [IQR 25%-75%] | Test Set (n = 490): %, Median [IQR 25%-75%] | P-Value |
|---|---|---|---|---|
| Low molecular weight heparin | No | 1131 (99.04) | 487 (99.39) | 0.680 |
| | Yes | 11 (0.96) | 3 (0.61) | 0.680 |
| Unfractioted heparin | No | 1140 (99.82) | 490.0 (100.00) | 0.877 |
| | Yes | 2 (0.18) | 0.0 (0.00) | 0.877 |
| Warfarin | No | 1122 (98.25) | 486 (99.18) | 0.225 |
| | Yes | 20 (1.75) | 4 (0.82) | 0.225 |
| Rivaroxaban | No | 1052 (92.12) | 453 (92.45) | 0.899 |
| | Yes | 90 (7.88) | 37 (7.55) | 0.899 |
| Sulfonylureas | No | 998 (87.39) | 416 (84.90) | 0.201 |
| | Yes | 144 (12.61) | 74 (15.10) | 0.201 |
| Glycosidase inhibitor | No | 998 (87.39) | 425 (86.73) | 0.777 |
| | Yes | 144 (12.61) | 65 (13.27) | 0.777 |
| Anti-infective treatment | No | 1032 (90.37) | 457 (93.27) | 0.072 |
| | Yes | 110 (9.63) | 33 (6.73) | 0.072 |
| Lipid medicine | No | 1100 (96.32) | 473 (96.53) | 0.951 |
| | Yes | 42 (3.68) | 17 (3.47) | 0.951 |
| Anti-platelet therapy during hospitalization | No | 1072 (93.87) | 466 (95.10) | 0.388 |
| | Yes | 70 (6.13) | 24 (4.90) | 0.388 |
| Anticoagulant therapy during hospitalization | No | 1008 (88.27) | 435 (88.78) | 0.833 |
| | Yes | 134 (11.73) | 55 (11.22) | 0.833 |
| Antilipidemic drugs during hospitalization | No | 1104 (96.67) | 474 (96.73) | 1.000 |
| | Yes | 38 (3.33) | 16 (3.27) | 1.000 |
| Antidiabetic treatment during hospitalization | No | 832 (72.85) | 351 (71.63) | 0.655 |
| | Yes | 310 (27.15) | 139 (28.37) | 0.655 |
| Chinese medicines during hospitalization | No | 1103 (96.58) | 482 (98.37) | 0.070 |
| | Yes | 39 (3.42) | 8 (1.63) | 0.070 |
| Intracranial artery stenosis | No | 1054 (92.29) | 459 (93.67) | 0.380 |
| | Yes | 88 (7.71) | 31 (6.33) | 0.380 |

## Model interpretation and SHAP Analysis

Ensuring the interpretability of predictive models, particularly in clinical settings, is crucial for their acceptance and application by healthcare professionals. To address this, our study employs SHAP methodology, enabling a transparent evaluation of how each variable influences the model's predictions. This approach affords a dual-layered interpretation: global insights, which elucidate the model's overall decision-making process, and local insights, which provide individualized explanations. The global interpretative framework is visualized through SHAP summary plots (Fig 3A and B), where the mean SHAP values of each feature are calculated and ranked. This hierarchy underscores the relative importance of predictors such as D-dimer levels, Prehospital medication, and Age, among others, in determining VTE risk. SHAP dependency plots further dissect the relationship between specific features and the prediction outcome, offering a granular understanding of feature impact.

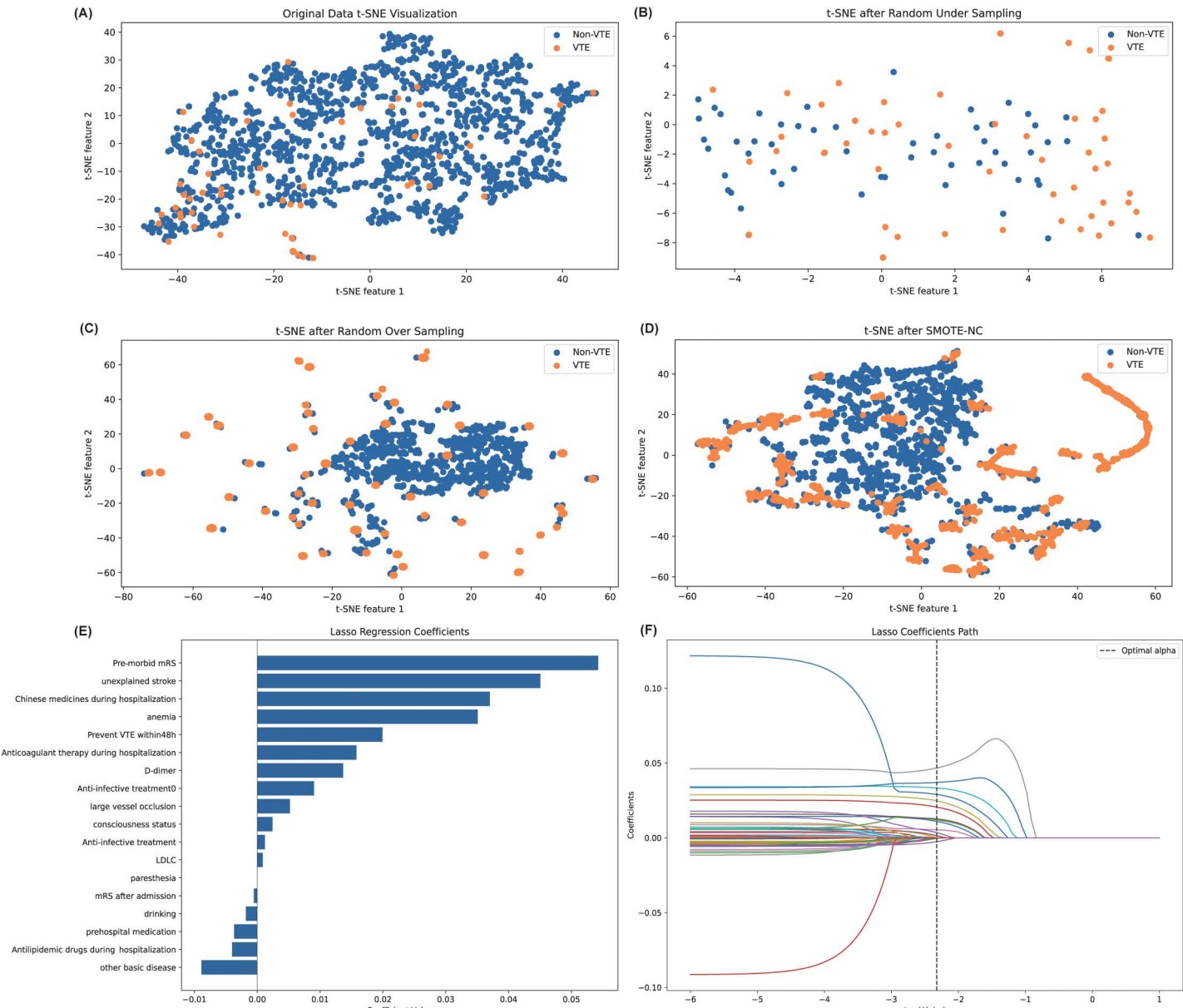

**Fig 1. Feature engineering.** (A) Original sample visualization using t-SNE shows an imbalanced outcome. (B) Random undersampling addresses imbalanced data by decreasing the majority class. (C) Random oversampling increases the minority class. (D) Synthetic Minority Over-sampling Technique with Nominal Continuous (SMOTE-NC) synthesizes data from the minority class. (E) Lasso Regression Coefficients: Indicates the influence of each feature as determined by the Lasso model. (F) Lasso coefficient path diagram.

Personalized risk assessments, as demonstrated in Fig 4A and B, highlight the model's ability to integrate individual patient data to predict VTE risk accurately. For instance, one patient was identified with a 99.7% VTE risk, with significant factors being premorbid mRS and age. Conversely, another patient presented a low risk of 1.1%, with premorbid mRS contributing negatively to VTE risk, indicating how diverse variables can influence individual risk profiles differently. Such analyses enable tailored patient care and informed risk management. Furthermore, Fig 4C reveals a nonlinear association between D-dimer levels and VTE risk, pinpointing a threshold beyond which VTE risk escalates significantly. This insight is critical

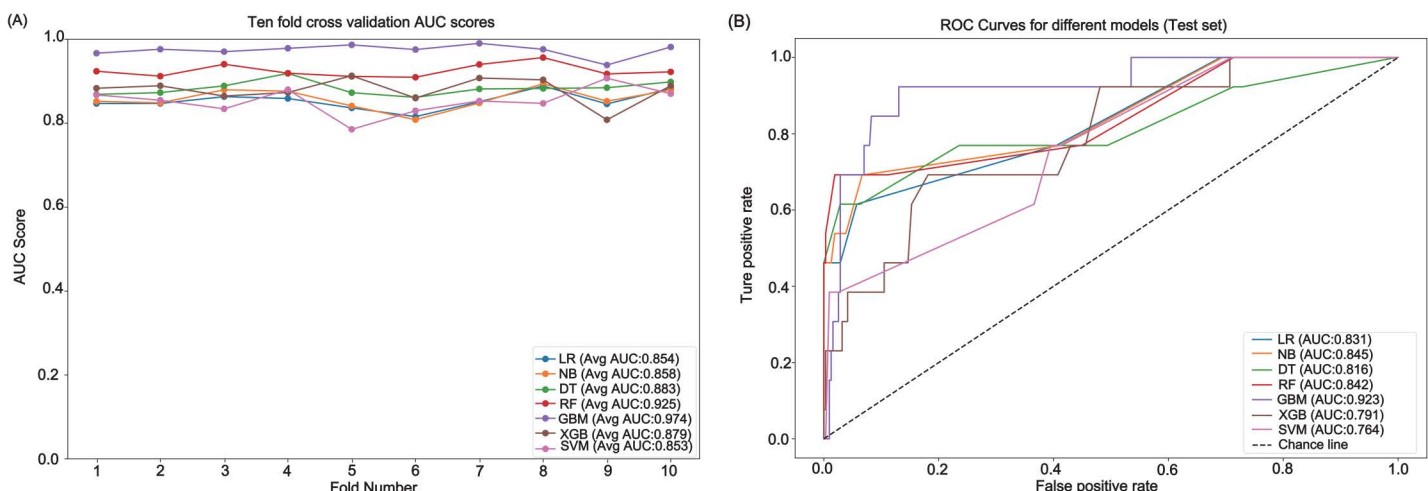

**Fig 2. Performance Comparison of Machine Learning Models.** (A) Ten-fold cross-validation AUC scores for various models, (B) ROC Curves for different models on the test set.

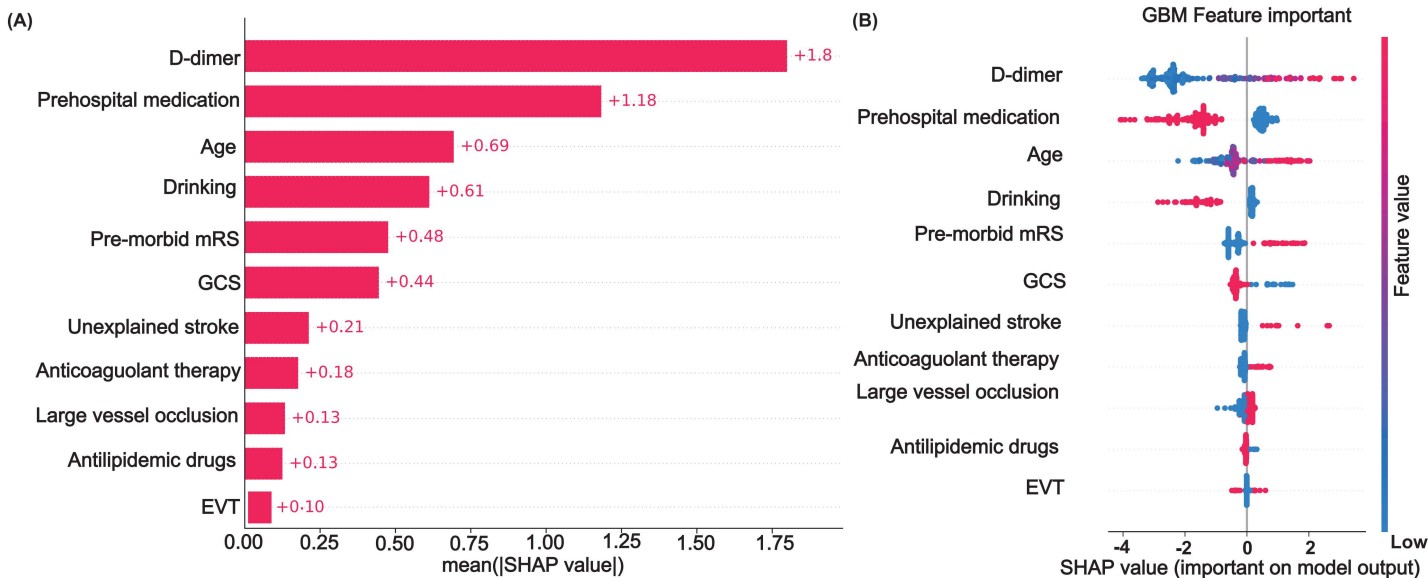

**Fig 3. Feature Impact Analysis.** (A) Mean SHAP values for predictors. (B) SHAP value distribution for GBM.

for identifying patients who might benefit from closer monitoring or preventive interventions. The SHAP dependency graph (Fig 4D) elaborates on the effect of individual variables across the patient cohort, providing a comprehensive overview of the model's predictive dynamics.

## Prognostic implications

The exemplary performance of the GBM model culminated in its integration into a user-friendly web application, designed to predict VTE risk in AIS patients based on the model's key variables. This digital tool, accessible at https://youlijiang236.shinyapps.io/myapp/,

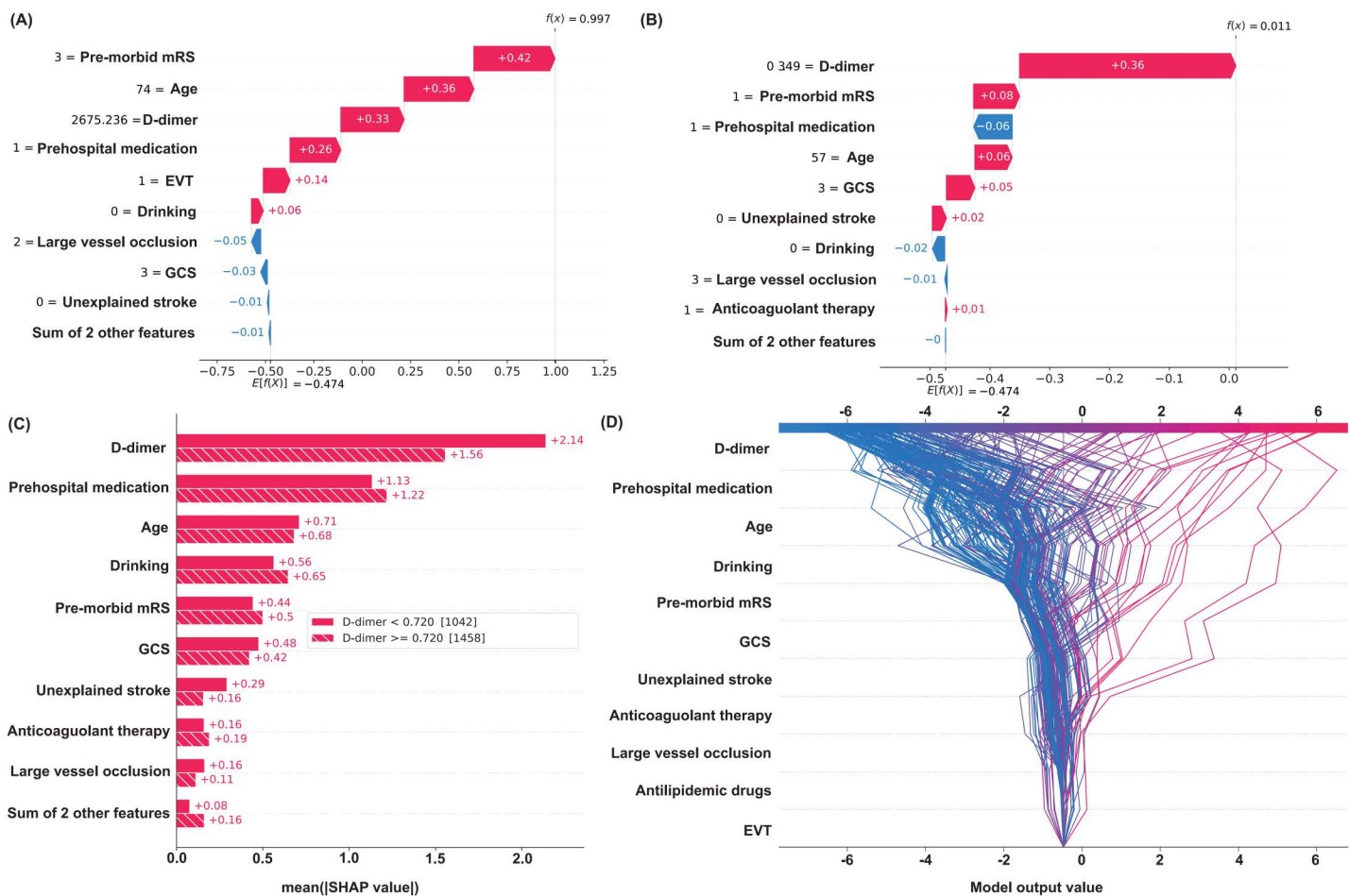

**Fig 4. SHAP Value Analysis for Model Prediction.** (A) Average impact on model output. (B) Impact of top features on a single prediction. (C) Aggregate SHAP values. (D) SHAP decision plot.

empowers clinicians to leverage our predictive model in real-time, facilitating personalized patient care and informed risk management strategies (Fig 5).

## Discussion

Our study developed a machine learning model to predict VTE risk in AIS patients, incorporating a broad range of predictive variables, including demographic, clinical, and laboratory data. A key innovation is the inclusion of previously underutilized factors, such as EVT and prehospital mRS scores, which contribute to a more comprehensive risk evaluation [19]. By leveraging the GBM algorithm, the model enhances both prediction accuracy and the ability to handle large datasets, making it highly practical for real-world applications [20].The integration of SHAP analysis provides crucial interpretability by showing how individual predictors influence VTE risk. Our model highlights critical predictors such as elevated D-dimer levels, pre-morbid mRS scores, and the presence of large vessel occlusion, which significantly contribute to VTE risk. These insights add value to the clinical application of the model by offering more precise and individualized risk assessments.

The GBM model developed in our study achieved an AUC of 0.923, demonstrating excellent discriminative ability to distinguish between patients at risk of developing VTE and those

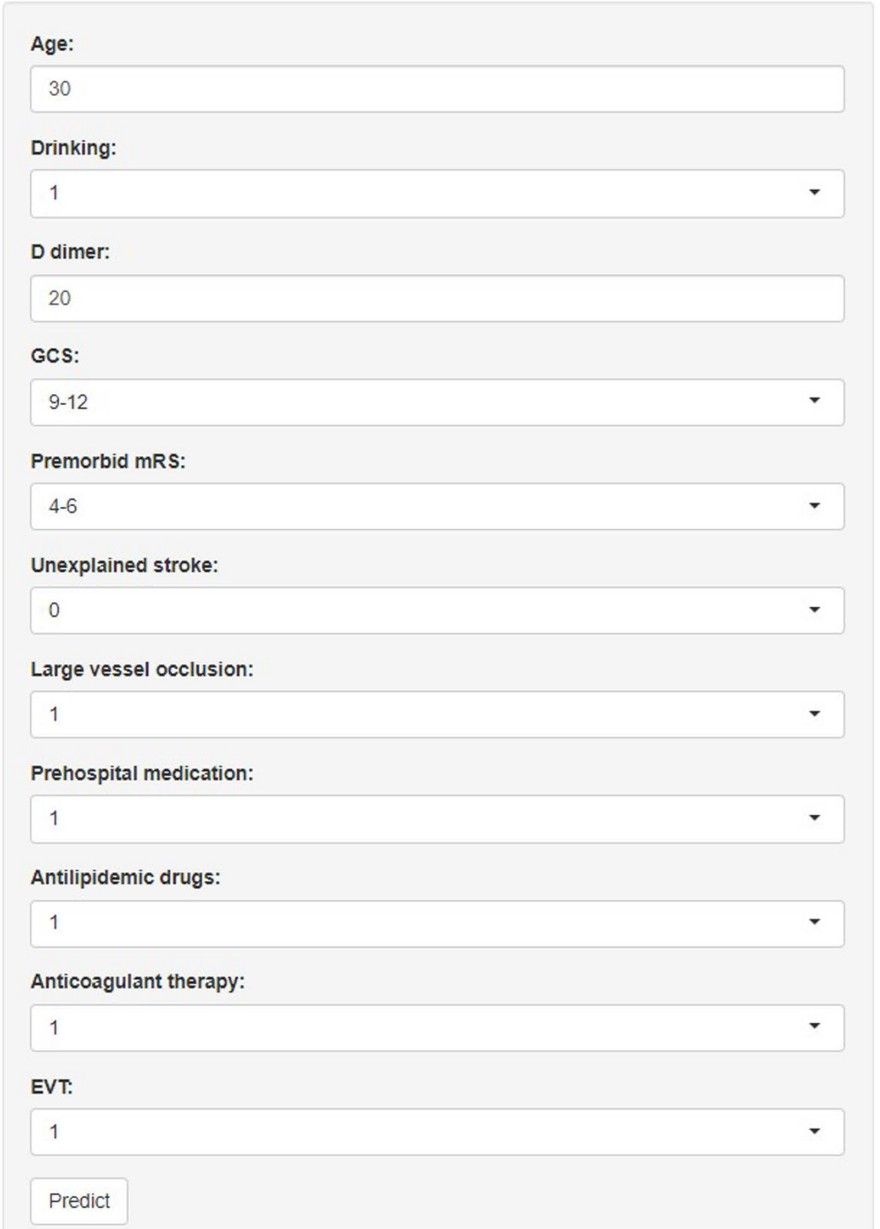

**Fig 5. VTE Risk Assessment Tool.**

who are not. The model's sensitivity (90.83%) and specificity (93.83%) further highlight its reliability in accurately identifying high-risk patients while minimizing false positives. This high level of performance offers a significant clinical utility, allowing for early interventions, such as initiating prophylactic anticoagulation therapy or mechanical thromboprophylaxis, which could reduce VTE incidence and associated morbidity [21]. Compared to other predictive models, our GBM model significantly surpasses traditional logistic regression approaches and recent machine learning models in both accuracy and clinical utility. Previous studies

using models like support vector machines for VTE prediction in medical inpatients achieved AUCs around 0.85, and random forest models applied to stroke patients reported even lower AUCs around 0.81 [22, 23]. Unlike these models, which focus solely on predictive performance, our GBM model excels by integrating interpretability with clinical relevance. Through the use of SHAP analysis, it not only delivers higher accuracy but also empowers clinicians by illuminating the specific contributions of each predictor, thus providing actionable insights. This dual strength of predictive power and interpretability positions our GBM model as a superior tool for VTE risk prediction in AIS patients, advancing both the science of prediction and practical clinical decision-making [24].

With the high sensitivity (90.83%) and specificity (93.83%) achieved by our model, the risk of false positives and false negatives is significantly minimized, reducing unnecessary interventions or delayed treatments [25]. The model's strong sensitivity ensures that high-risk patients are correctly identified, while its high specificity reduces the likelihood of inappropriate anticoagulation therapy and the associated risks of bleeding [26]. Despite these positive results, clinical judgment remains essential to manage any uncertainty in borderline cases or when the predicted risk is ambiguous. Establishing protocols for high-risk predictions and regularly monitoring model performance will further enhance patient safety and optimize treatment outcomes [27]. The real-time integration of the model into electronic health records (EHRs) offers substantial improvements in VTE risk stratification for AIS patients. This allows healthcare providers to administer prophylactic interventions, such as low molecular weight heparin, to high-risk patients early while minimizing unnecessary treatment for low-risk patients, thus mitigating the risk of adverse effects like bleeding [28]. Additionally, SHAP-based interpretability enhances clinicians' understanding of key risk factors, enabling more precise, patient-specific treatment plans and fostering informed discussions within multidisciplinary teams [15].

Our machine learning model offers clear advantages over traditional clinical risk scores like the Caprini and Wells scores, which are widely used but not tailored for AIS patients. The Caprini score was developed for surgical populations and assigns equal weight to various risk factors, which may not accurately reflect the unique risk profiles in stroke patients [7]. Similarly, the Wells score focuses on existing VTE probability rather than predicting future risk in hospitalized patients [29]. In contrast, our model incorporates a broad range of stroke-specific variables and uses advanced algorithms to provide a more nuanced and accurate risk assessment, allowing for better identification of high-risk patients who might be overlooked by traditional scores [30]. Additionally, our methodological approach combines LASSO and stepwise logistic regression to optimize feature selection, while addressing multicollinearity through LASSO's regularization properties [31]. PCA further condenses information from correlated variables, ensuring that predictive power is not compromised [32]. This synergy between LASSO and PCA enhances model stability, reduces the risk of overfitting, and significantly improves generalizability. By leveraging these advanced techniques, our model maintains accuracy across a wide range of patient data, demonstrating a critical advancement in predictive modeling [33].

Our GBM model demonstrated superior predictive performance, with an AUC of 0.923, significantly outperforming both traditional and state-of-the-art machine learning models specifically designed for lower extremity deep vein thrombosis prediction in stroke patients. Prior models have shown AUCs ranging from 0.724 to 0.907, depending on the cohort and methodology [34, 35]. However, what sets our model apart is its ability to maintain high predictive accuracy across diverse patient populations, while simultaneously offering clinicians enhanced interpretability through SHAP analysis. This interpretability is crucial in tailoring patient-specific interventions, allowing healthcare providers

to not only predict risk but also understand the key factors driving that risk [36]. This unique combination of robust performance and clinically meaningful insights underscores the value of our GBM model in elevating VTE risk assessment in AIS patients. Moving forward, the practical application of our model is poised to revolutionize VTE risk screening in AIS patients through its integration into clinical workflows. This server-based web calculator allows healthcare providers to input patient-specific data and receive real-time VTE risk predictions and more accurately identify high-risk patients, facilitating early intervention strategies such as anticoagulation or mechanical thromboprophylaxis. The model's predictions can also inform multidisciplinary discussions, improving treatment planning and resource allocation [37]. By integrating our model into routine practice, clinicians can better manage VTE risk in AIS patients, while SHAP analysis continues to provide valuable interpretability for individual risk profiles [38]. This interpretability supports more personalized care and allows for more informed decisions regarding treatment modalities, enhancing overall patient outcomes.

One limitation of our study is the lack of external validation, which may limit the generalizability of the model to broader populations and clinical settings. Although our internal testing demonstrated strong predictive performance, applying the model to independent datasets would further confirm its reliability and applicability across diverse patient groups. Additionally, while SHAP values were utilized for model interpretability, the current web-based calculator lacks visual representations to illustrate the contributions of specific indicators to VTE risk. Future updates will aim to incorporate these SHAP-based visualizations, enhancing both the user experience and the clinical utility of the tool.

## Conclusion

Our study enhances VTE risk prediction in acute ischemic stroke patients through the use of a GBM algorithm, offering a more precise and tailored assessment tool compared to traditional methods. The model's integration of SHAP values improves interpretability, allowing clinicians to make informed decisions on personalized treatment plans. This model has the potential to be integrated into clinical workflows and decision-support systems, thereby improving early detection and management of VTE. Future research should explore its practical implementation to optimize clinical efficiency and patient outcomes in diverse healthcare settings.

## Supporting information

**S1 Table.  Raw data integrity investigation.**
(DOCX)

**S2 Table.  Stepwise forward logistic regression multivariate analysis.**
(DOCX)

**S3 File.  The predictive performance on test set.**
(XLSX)

**S4 File.   Precision-Recall Curve.**
(PDF)

## Acknowledgments

We would like to thank the doctors and nurses at the Department of Neurology, Longhua District People's Hospital, Shenzhen for their help and support in this study.

## Author contributions

**Data curation:** Rong Li, Guisu Li.

**Formal analysis:** Youli Jiang, Ao Li, Guisu Li.

**Funding acquisition:** Qingshi Zhao.

**Project administration:** Yanfeng Li, Qingshi Zhao.

**Resources:** Guisu Li.

**Software:** Youli Jiang.

**Supervision:** Yanfeng Li, Qingshi Zhao.

**Validation:** Youli Jiang.

**Visualization:** Youli Jiang.

**Writing – original draft:** Youli Jiang.

**Writing – review & editing:** Youli Jiang, Ao Li, Zhihuan Li, Qingshi Zhao, Guisu Li.

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
