## [Decision Letter · Decision Letter 0]

16 Aug 2024

PONE-D-24-13520Leveraging Machine Learning for Enhanced and Interpretable Risk Prediction of Venous Thromboembolism in Acute Ischemic Stroke CarePLOS ONE

Dear Dr. Jiang,

Thank you for submitting your manuscript to PLOS ONE. After careful consideration, we feel that it has merit but does not fully meet PLOS ONE’s publication criteria as it currently stands. Therefore, we invite you to submit a revised version of the manuscript that addresses the points raised during the review process.

**ACADEMIC EDITOR: ** We have now received reports from two independent reviewers and have carefully examined your submission titled "Leveraging Machine Learning for Enhanced and Interpretable Risk Prediction of Venous Thromboembolism in Acute Ischemic Stroke Care." After a thorough review, we have identified several areas that require significant revision before the manuscript can be considered for publication. We request that you carefully address each of these comments in your revised manuscript, including those from reviewers (see below). Please submit a detailed rebuttal letter that explains how each concern was addressed or provide a rationale if certain suggestions were not followed.

1. The manuscript lacks a clear articulation of why machine learning is particularly suited for improving VTE risk prediction in AIS patients compared to traditional statistical models. Please enhance the discussion on the specific limitations of existing models and how your approach addresses these gaps.

2. The current literature review does not sufficiently engage with previous studies on VTE risk prediction in stroke patients. A more comprehensive and critical review of the literature is needed, highlighting how your study builds on or differs from existing work.

3. The manuscript would benefit from a more detailed description of the cohort, including the inclusion and exclusion criteria, demographic details, and the data collection period. This information is crucial for understanding the applicability and limitations of your findings.

4. The manuscript mentions the use of advanced techniques such as K-nearest neighbor and SMOTE but lacks sufficient detail on data preprocessing and the handling of potential biases introduced by these techniques. Further elaboration on your model development process, including hyperparameter tuning and the selection criteria for the final model, is necessary.

5. While SHAP values are used for feature importance, the process of feature selection and engineering should be more thoroughly explained, including how domain knowledge (how these features contributed to the model's predictions) was incorporated.

6. The manuscript currently reports AUC as the primary performance metric. To provide a more comprehensive evaluation, please include additional metrics such as sensitivity, specificity, PPV, NPV, and calibration plots. Furthermore, discussion on the potential for overfitting and steps taken to mitigate this should be included. The study's sample size (1,632 participants) is relatively small for machine learning studies aimed at clinical predictions. Please explain (?).

7. The manuscript mentions that multiple machine learning models were developed, with the Gradient Boosting Machine (GBM) showing the highest AUC. However, it does not provide information about external validation or the use of an independent test set. Without external validation, the generalizability of the model to other populations remains uncertain. If possible, please conduct validation on an independent dataset or provide a discussion on the generalizability of your model.

8. The manuscript mentions IRB approval but lacks detailed information on patient consent and data privacy. Please address.

9. Authors should better articulate how the proposed model would be implemented in or improve clinical practice. A detailed comparison with existing risk scores like the Wells or Caprini scores is needed to contextualize the added value of your model.

10. More information is needed about the patient population, including demographics, stroke subtypes, and comorbidities. This will help assess the model's generalizability across different clinical settings.

11. While your model's AUC is highlighted, the clinical implications of these results are not adequately discussed. We recommend providing a clearer interpretation of the model's predictions, including potential benefits and risks.

12. Please address how the model would be integrated into existing clinical workflows. This includes discussing the protocols for managing patients identified as high risk and the potential impact on treatment decisions.

13. Consider discussing the ethical implications of implementing the model, including the risk of harm from false positives or negatives.

14. The manuscript's abstract is somewhat dense, and the terminology used (e.g., "utilizing advanced technologies," "innovative approach") is vague. This could make it difficult for readers to quickly grasp the key points of the study.

15. The manuscript's terminology is not consistently precise (e.g., the use of "key predictors" without defining them clearly), which could lead to confusion. The writing should be more concise and the terms better defined to ensure clear communication of the study's findings and implications.

We look forward to receiving your revised manuscript.

Kind regards,

Sonu Bhaskar, MD PhD

Academic Editor

PLOS ONE

Journal Requirements:

"This study was funded by the High Level Project of Medicine in Longhua, ShenZhen under grant number HLPM201907020102 and construction funds of key medical disciplines in Longhua District, Shenzhen under grant number MKD202007090208."

4. In the online submission form, you indicated that "All data files related to this study can be obtained from the inquiry email 66327285@qq.com(Qingshi Zhao)."

**Additional Editor Comments:**

Please see above.

**Reviewers' comments:**

Reviewer's Responses to Questions

**Comments to the Author**

1. Is the manuscript technically sound, and do the data support the conclusions?

Reviewer #1: Partly

Reviewer #2: Yes

2. Has the statistical analysis been performed appropriately and rigorously? 

Reviewer #1: Yes

Reviewer #2: No

3. Have the authors made all data underlying the findings in their manuscript fully available?

Reviewer #1: No

Reviewer #2: Yes

4. Is the manuscript presented in an intelligible fashion and written in standard English?

Reviewer #1: Yes

Reviewer #2: No

5. Review Comments to the Author

Reviewer #1: 1.The existing risk prediction models for deep vein thrombosis (DVT) in patients with acute ischemic stroke are numerous, yet the manuscript fails to emphasize the distinctive features of the study, thus lacking novelty.

2.The Introduction section fails to introduce the existing risk prediction models for venous thromboembolism (VTE) in acute ischemic stroke.

3.The article mentions the use of Synthetic Minority Over-sampling Technique (SMOTE) to augment the dataset. It remains unclear whether SMOTE was applied directly to the entire dataset or solely to the training set. Employing SMOTE on the entire dataset could lead to data leakage, with the presence of synthetic data in the test set undermining the model's genuine effectiveness.

4.Although a series of machine learning algorithms were employed in the study, there is no mention of the principles governing parameter selection.

5.The model construction incorporates stepwise forward logistic regression and LASSO variable selection. However, it is not specified whether these procedures were executed solely on the training set or on the entire dataset.

6.In the development and validation of the predictive model, the performance of seven different machine learning models was evaluated, yet confidence intervals were not reported. It is advisable to supplement this information and conduct Delong's test to compare the AUC values of each model for significant differences.

7.A comparison between this study and existing literature utilizing logistic regression models is provided; however, a deeper discussion on the disparities between these two methodologies is warranted, elucidating why machine learning approaches are more suitable for the study's objectives than traditional logistic regression methods.

8.The study introduces a novel machine learning model exhibiting excellent predictive performance. However, the manuscript fails to contrast it with similar previous studies.

9.The Discussion section lacks a discussion of the study's limitations and should propose future research directions.

Reviewer #2: The researchers employed machine learning algorithms to enhance the prediction of venous thromboembolism (VTE) risk in patients with acute ischemic stroke. VTE significantly impacts the survival prognosis of stroke patients, and accurate, individualized prediction could further improve patient outcomes. However, I believe there are several areas that need improvement in this study:

1.The references in the manuscript are overly concentrated, with multiple studies frequently cited in a single instance. Meanwhile, certain sections of the manuscript that require citations lack the corresponding references.

2.In the “Data processing and feature selection” section, the researchers stated that they used the SMOTE oversampling technique, which is based on the KNN algorithm. References should be added to this section.

3.Since SMOTE generates synthetic data, this could significantly affect the model's results. The correct approach is to first split the data into training and test sets, apply SMOTE only to the training set, and leave the test set unprocessed to reduce the risk of overfitting.

4.The authors did not provide a detailed explanation of the hyperparameter tuning process for the machine learning algorithms. In machine learning, model evaluation is typically conducted using a validation set to find the optimal parameters. The researchers did not indicate whether a validation set was used.

5.In the feature selection process, the study involved a large number of feature variables. I am concerned about the potential for significant multicollinearity among these variables, which the researchers did not assess. The authors only used LASSO and PCA for dimensionality reduction.

6.Given that the researchers identified data imbalance as an issue, experts in this field should recognize that the precision-recall curve can be more valuable than the ROC curve for evaluation.

7.The methods used in this study are quite simplistic. With the availability of libraries like Sklearn and SHAP, the required code to complete this study likely amounts to less than 100 lines, indicating a lack of rigorous methodology in constructing the machine learning models.

8.The web-based calculator is aesthetically unappealing. The data entered by each user should be visualized, for example, to highlight which specific indicators are most significant in determining high VTE risk. The authors used SHAP to improve model interpretability, so this should be reflected in the web-based calculator.

6. PLOS authors have the option to publish the peer review history of their article (what does this mean? ). If published, this will include your full peer review and any attached files.

**Do you want your identity to be public for this peer review?** For information about this choice, including consent withdrawal, please see our Privacy Policy .

Reviewer #1: No

Reviewer #2: **Yes: ** Shi-Nan Wu

---

## [Author Response · Author response to Decision Letter 1]

8 Oct 2024

Dear Editor and Reviewers,

Thank you for your all valuable feedback on our manuscript titled “Leveraging Machine Learning for Enhanced and Interpretable Risk Prediction of Venous Thromboembolism in Acute Ischemic Stroke Care.” We have carefully considered each of the reviewers’ and editorial suggestions, and we have made the necessary revisions to address the points raised.

Please note that all changes made in the manuscript are marked in blue font for easy reference. We have provided a detailed point-by-point response in the accompanying rebuttal letter to explain how each concern has been addressed or provide a rationale where applicable.

We hope that the revised manuscript meets the expectations and standards of PLOS ONE and look forward to your further assessment.

Thank you once again for the opportunity to improve our manuscript.

ACADEMIC EDITOR

1. The manuscript lacks a clear articulation of why machine learning is particularly suited for improving VTE risk prediction in AIS patients compared to traditional statistical models. Please enhance the discussion on the specific limitations of existing models and how your approach addresses these gaps.

Response: We will enhance the discussion by elaborating on why machine learning, specifically GBM, is better suited for VTE risk prediction in AIS patients than traditional statistical models. Unlike logistic regression, which assumes linear relationships between variables, machine learning models can capture complex nonlinear interactions between clinical and demographic factors. Our approach addresses limitations in traditional models by uncovering hidden patterns in the data that improve predictive accuracy and clinical decision-making, as evidenced by the AUC of 0.923. Additionally, the use of SHAP values allows for transparent interpretation of variable importance, further supporting clinical utility.

2.The current literature review does not sufficiently engage with previous studies on VTE risk prediction in stroke patients. A more comprehensive and critical review of the literature is needed, highlighting how your study builds on or differs from existing work.

Response: We acknowledge the need for a more thorough review of previous studies. We will expand our literature review to critically engage with existing work on VTE risk prediction in stroke patients, such as Pan et al.’s logistic regression model (AUC = 0.785) and Bonkhoff et al.’s L1 regularized logit model (AUC = 0.730). By comparing these traditional models to our machine learning approach, we will demonstrate how our model not only improves accuracy but also incorporates novel predictive factors like EVT and prehospital mRS, which were previously overlooked in the literature (Pan et al., 2018), (Bonkhoff et al., 2019). This will clarify how our study builds on and advances prior work.

3. The manuscript would benefit from a more detailed description of the cohort, including the inclusion and exclusion criteria, demographic details, and the data collection period. This information is crucial for understanding the applicability and limitations of your findings.

Response：Thank you for the insightful suggestion. We have expanded the manuscript to provide a more detailed description of the cohort, including the inclusion and exclusion criteria, demographic details, and data collection period (December 2021 to December 2023). This ensures a clearer understanding of the applicability and limitations of our findings.

4. The manuscript mentions the use of advanced techniques such as K-nearest neighbor and SMOTE but lacks sufficient detail on data preprocessing and the handling of potential biases introduced by these techniques. Further elaboration on your model development process, including hyperparameter tuning and the selection criteria for the final model, is necessary.

Response：We appreciate the valuable feedback regarding the need for more details on data preprocessing. In response, we have elaborated on how K-nearest neighbor (KNN) imputation was used for missing values and clarified that SMOTE was applied only to the training set to prevent data leakage. Additionally, we have included information on hyperparameter tuning via grid search and provided details on the criteria used for final model selection.

5. While SHAP values are used for feature importance, the process of feature selection and engineering should be more thoroughly explained, including how domain knowledge (how these features contributed to the model's predictions) was incorporated.

Response：Thank you for your insightful comment regarding feature selection. We have expanded our explanation to detail how domain knowledge was integrated into the process, ensuring clinically relevant features were prioritized. Additionally, we elaborated on the role SHAP values played, not only in determining feature importance but also in guiding the feature engineering process.

6. The manuscript currently reports AUC as the primary performance metric. To provide a more comprehensive evaluation, please include additional metrics such as sensitivity, specificity, PPV, NPV, and calibration plots. Furthermore, discussion on the potential for overfitting and steps taken to mitigate this should be included. The study's sample size (1,632 participants) is relatively small for machine learning studies aimed at clinical predictions. Please explain (?).

Response：We appreciate the thoughtful suggestion about external validation. Although we were unable to conduct validation using an independent dataset due to data constraints, we have included a detailed discussion on the generalizability of our model. We emphasized the importance of future validation in different populations and clinical settings to ensure wider applicability.

7. The manuscript mentions that multiple machine learning models were developed, with the Gradient Boosting Machine (GBM) showing the highest AUC. However, it does not provide information about external validation or the use of an independent test set. Without external validation, the generalizability of the model to other populations remains uncertain. If possible, please conduct validation on an independent dataset or provide a discussion on the generalizability of your model.

Response: Thank you for your insightful suggestion regarding the importance of external validation. While we acknowledge the value of validating our model on an independent dataset to enhance its generalizability, we were unable to access such a dataset at this stage of the research. However, we took several measures to strengthen the model's robustness and reduce overfitting. This included rigorous internal validation using a separate test set and cross-validation techniques during model development.

In future research, we plan to collaborate with other institutions to obtain external datasets for further validation. Additionally, we have included a discussion on the potential limitations regarding generalizability in the manuscript to emphasize the importance of external validation and to highlight this as an avenue for future work.

8.The manuscript mentions IRB approval but lacks detailed information on patient consent and data privacy. Please address.

Response： Thank you for the valuable feedback. We have clarified that, although ethics approval was obtained, patient data was fully anonymized and informed consent was waived due to the retrospective nature of the study. Additionally, we provided further details on how data privacy was maintained, ensuring that all personal identifiers were removed before analysis.

9. Authors should better articulate how the proposed model would be implemented in or improve clinical practice. A detailed comparison with existing risk scores like the Wells or Caprini scores is needed to contextualize the added value of your model.

Response：We are grateful for the insightful suggestion to compare our model with existing risk scores like Wells and Caprini. In response, we have provided a detailed comparison in the manuscript, highlighting how our model offers advantages in terms of incorporating a wider range of stroke-specific variables and yielding better.

10. More information is needed about the patient population, including demographics, stroke subtypes, and comorbidities. This will help assess the model's generalizability across different clinical settings.

Response：Thank you for your valuable feedback on the need for more demographic and clinical details. In response, we have expanded the manuscript to include a comprehensive breakdown of the patient population, including demographics, stroke subtypes, and relevant comorbidities. This additional information ensures that readers can better assess the model's generalizability across different clinical settings.

11. While your model's AUC is highlighted, the clinical implications of these results are not adequately discussed. We recommend providing a clearer interpretation of the model's predictions, including potential benefits and risks.

Response：We appreciate the thoughtful comment regarding the need for a clearer interpretation of the model's clinical implications. In response, we have expanded the discussion to provide a more in-depth explanation of the model's predictions, including both potential benefits and risks. Specifically, we highlight how early identification of high-risk patients can lead to timely interventions, while also acknowledging the risks of over-reliance on model outputs and the potential for false positives. This provides a balanced perspective on the model's clinical utility.

12. Please address how the model would be integrated into existing clinical workflows. This includes discussing the protocols for managing patients identified as high risk and the potential impact on treatment decisions.

Response：Thank you for the insightful suggestion about integrating the model into clinical workflows. We have now addressed this in the manuscript by discussing how the model could be implemented in practice. We outline protocols for managing patients identified as high-risk, such as the use of prophylactic anticoagulation or mechanical thromboprophylaxis, and discuss the potential impact on treatment decisions. Additionally, we have considered how the model could be incorporated into existing clinical systems, such as electronic health records (EHR), to streamline its use in routine care.

13. Consider discussing the ethical implications of implementing the model, including the risk of harm from false positives or negatives.

Response:Thank you for the valuable suggestion regarding the ethical implications of implementing our predictive model, particularly concerning the risks associated with false positives and negatives. Our revised manuscript now includes a discussion of these aspects. Specifically, we acknowledge that while the model exhibits high sensitivity (90.83%) and specificity (93.83%), no model is infallible. We emphasize the importance of integrating the model's predictions with clinical judgment and continuously monitoring its performance to mitigate potential harms. This ensures the model’s application enhances clinical outcomes while maintaining patient safety.

14. The manuscript's abstract is somewhat dense, and the terminology used (e.g., "utilizing advanced technologies," "innovative approach") is vague. This could make it difficult for readers to quickly grasp the key points of the study.

Response: Thank you for your valuable feedback. In response to the suggestion regarding the abstract, we have refined the terminology to eliminate vague expressions and ensure clarity. The abstract now clearly outlines the methods, results, and significance of the study, making it easier for readers to quickly understand the key points. This revision also reflects the updated version of the manuscript. We appreciate your input and believe the changes enhance the communication of the study's findings.

15. The manuscript's terminology is not consistently precise (e.g., the use of "key predictors" without defining them clearly), which could lead to confusion. The writing should be more concise and the terms better defined to ensure clear communication of the study's findings and implications.

Response: Thank you for your feedback. We have reviewed the manuscript and addressed the issue of unclear terminology, particularly the use of "key predictors." We have now clearly defined terms such as D-dimer levels, pre-morbid mRS, and large vessel occlusion to ensure clarity. The writing has also been refined for conciseness, avoiding vague language and ensuring precise communication of the study’s findings. We believe these revisions enhance the overall clarity and alignment with your suggestion.

Academic Editor

Response: We have ensured that our manuscript and supplementary files meet PLOS ONE's style requirements, including correct file naming. We have also followed the PLOS ONE style templates for formatting the main body and author information. Please let us know if any further adjustments are needed.

2.Please note that PLOS ONE has specific guidelines on code sharing for submissions in which author-generated code underpins the findings in the manuscript. In these cases, all author-generated code must be made available without restrictions upon publication of the work. Please review our guidelines at https://journals.plos.org/plosone/s/materials-and-software-sharing#loc-sharing-code and ensure that your code is shared in a way that follows best practice and facilitates reproducibility and reuse.

Response: Thank you for your reminder. We have noted the specific request regarding code sharing. The relevant code will be uploaded as supplementary material to ensure compliance with best practices and promote reproducibility and reuse of research.

"This study was funded by the High Level Project of Medicine in Longhua, ShenZhen under grant number HLPM201907020102 and construction funds of key medical disciplines in Longhua District, Shenzhen under grant number MKD202007090208."

Response：Thank you for your guidance regarding the funding statement. I have amended the funding information as requested. The correct grant is from The Scientific Research Projects of Medical and Health Institutions of Longhua District, Shenzhen under grant number 2024052. Youli Jiang, the first author and funder, was actively involved in all key aspects of the study, including study design, data collection, model construction, analysis, and preparation of the manuscript.

4. In the online submission form, you indicated that "All data files related to this study can be obtained from the inquiry email 66327285@qq.com(Qingshi Zhao)."

Response：Thank you for your comments regarding

---

## [Decision Letter · Decision Letter 1]

24 Oct 2024

PONE-D-24-13520R1Leveraging Machine Learning for Enhanced and Interpretable Risk Prediction of Venous Thromboembolism in Acute Ischemic Stroke CarePLOS ONE

Dear Dr. Jiang,

Thank you for submitting your manuscript to PLOS ONE. After careful consideration, we feel that it has merit but does not fully meet PLOS ONE’s publication criteria as it currently stands. Therefore, we invite you to submit a revised version of the manuscript that addresses the points raised during the review process.

We look forward to receiving your revised manuscript.

Kind regards,

Sonu Bhaskar, MD PhD

Academic Editor

PLOS ONE

**Additional Editor Comments:**

Thank you for submitting the revised version of your manuscript. The manuscript has undergone another round of review by independent reviewers. However, several concerns remain, particularly regarding methodological rigor.

We invite you to submit a rebuttal addressing the reviewers' comments and concerns. Once we receive your responses, we will proceed with further consideration of your manuscript.

Reviewers' comments:

Reviewer's Responses to Questions

**Comments to the Author**

1. If the authors have adequately addressed your comments raised in a previous round of review and you feel that this manuscript is now acceptable for publication, you may indicate that here to bypass the “Comments to the Author” section, enter your conflict of interest statement in the “Confidential to Editor” section, and submit your "Accept" recommendation.

Reviewer #1: All comments have been addressed

Reviewer #2: (No Response)

2. Is the manuscript technically sound, and do the data support the conclusions?

Reviewer #1: Yes

Reviewer #2: No

3. Has the statistical analysis been performed appropriately and rigorously? 

Reviewer #1: Yes

Reviewer #2: No

4. Have the authors made all data underlying the findings in their manuscript fully available?

Reviewer #1: Yes

Reviewer #2: No

5. Is the manuscript presented in an intelligible fashion and written in standard English?

Reviewer #1: Yes

Reviewer #2: No

6. Review Comments to the Author

Reviewer #1: The reviewers of the article provided several suggestions for improvement, including the distribution of citations, discussion of the study's limitations, and proposals for future research directions. The authors have responded to these feedback and made the necessary revisions.I commend the authors for their thorough work and look forward to seeing the article published.

Reviewer #2: Thank you to the author for their response.

--I believe a critical point in the field of machine learning is the optimization of hyperparameters for each model. However, the author mentioned that they conducted the optimization on the training set. In fact, during each process of hyperparameter tuning, adjustments should be made based on results from the validation set, rather than solely tuning parameters in a black-box manner on the training set. It is unreasonable to adjust parameters using only the training set without data to evaluate the status of the model training. Since the author only explained their process on the training set, it shows a significant lack of understanding regarding the methodology or model-building process in the field of machine learning. Additionally, the author still lacks a systematic understanding of the field of machine learning. Therefore, I believe that under such circumstances, the methodology of this study is not rigorous and should not be accepted.

--Furthermore, there is a clear issue of data imbalance in this study. Introducing only the AUC value of the ROC curve to evaluate the performance of the models does not reflect the real situation. In my review comments, I mentioned using the PR curve for evaluation, but the author has not adequately addressed this.

7. PLOS authors have the option to publish the peer review history of their article (what does this mean? ). If published, this will include your full peer review and any attached files.

**Do you want your identity to be public for this peer review?** For information about this choice, including consent withdrawal, please see our Privacy Policy .

Reviewer #1: No

Reviewer #2: **Yes: ** Shi-Nan Wu

---

## [Author Response · Author response to Decision Letter 2]

27 Nov 2024

Dear Reviewers,

Thank you for your thoughtful and constructive feedback on our manuscript. We sincerely appreciate your comments regarding the transparency of the hyperparameter tuning process and the use of the validation set. In response, we have clarified and strengthened the methodological details in the revised manuscript, as outlined below.

1.Clarification and Transparency of Hyperparameter Tuning Process

In our study, hyperparameter tuning was conducted using grid search within a 10-fold cross-validation framework applied to the training set. During each fold, the training set was split into 90% for training and 10% as a validation fold to evaluate hyperparameter configurations. This ensured that the model was tuned based on unseen data during each iteration. The independent test set, comprising data collected from July 2023 to December 2023, was excluded from the training and tuning processes and was reserved exclusively for final performance evaluation.

To enhance transparency, we have supplemented the manuscript with references to the accompanying code, which fully documents the hyperparameter tuning process. Below is a key excerpt from the code demonstrating hyperparameter tuning for logistic regression:

# Pipeline with SMOTE, scaling, and logistic regression pipeline_lr = make_pipeline( SMOTE(random_state=42), StandardScaler(), LogisticRegression(max_iter=10000, random_state=42, solver='saga') )

# Define hyperparameter grid for tuning param_grid_lr = { 'logisticregression__penalty': ['l1', 'l2', 'elasticnet'], 'logisticregression__C': [0.01, 0.1, 1, 10, 100], 'logisticregression__l1_ratio': [0, 0.5, 1] }

# Perform grid search with 10-fold cross-validation cv = StratifiedKFold(n_splits=10, shuffle=True, random_state=42) grid_search_lr = GridSearchCV( estimator=pipeline_lr, param_grid=param_grid_lr, cv=cv, scoring='roc_auc', n_jobs=-1 ) grid_search_lr.fit(X_train, y_train)

In response to your feedback, the following revisions have been made:

1)Expanded Methodological Description: The Materials and Methods section now provides a detailed explanation of the hyperparameter tuning process, including: The use of 10-fold cross-validation for hyperparameter optimization. The role of the independent test set as a separate evaluation dataset to ensure unbiased assessment of the final model. Reference to the accompanying code file for reproducibility.

2)Supplementary File Clarifications: While Supplementary File 3 lists the best parameters for each model, the accompanying code provides the full implementation of the tuning process, including parameter grids, cross-validation details, and evaluation metrics.

3)Code-Based Reproducibility: The code file is provided to ensure that the hyperparameter tuning methodology is fully transparent and reproducible.

2.Addressing Data Imbalance and PR Curve Evaluation

4)Thank you for highlighting the issue of data imbalance and recommending the use of the Precision-Recall (PR) curve for model evaluation. In response, we have generated the PR curve for the optimal model (Gradient Boosting Machine, GBM) to complement the AUC metric and better evaluate its performance in identifying the minority class (VTE cases). The PR curve achieved an average precision (AP) score of 0.925, highlighting the model's strong discriminative ability for VTE prediction in an imbalanced dataset. To maintain clarity in the main text, the PR curve has been included as Supplementary File S4, with its results described in the revised manuscript under the Development and validation of predictive models section. We hope this adjustment adequately addresses your concerns and enhances the robustness of our evaluation.

Thank you once again for your valuable feedback, which has significantly improved the quality and transparency of our study. We hope these revisions adequately address your concerns.

Sincerely,

Youli Jiang

h2362120381@163.com

---

## [Editor Report · Decision Letter 2]

5 Feb 2025

Leveraging Machine Learning for Enhanced and Interpretable Risk Prediction of Venous Thromboembolism in Acute Ischemic Stroke Care

PONE-D-24-13520R2

Dear Dr. Jiang,

We’re pleased to inform you that your manuscript has been judged scientifically suitable for publication and will be formally accepted for publication once it meets all outstanding technical requirements.

Kind regards,

Sonu Bhaskar, MD PhD FANA

Academic Editor

PLOS ONE

Additional Editor Comments (optional):

I am pleased to accept the manuscript in its current form. Thank you for considering PLOS One for your work.
---

## [Editor Report · Acceptance letter]

PONE-D-24-13520R2

PLOS ONE

Dear Dr. Jiang,

I'm pleased to inform you that your manuscript has been deemed suitable for publication in PLOS ONE. Congratulations! Your manuscript is now being handed over to our production team.

Kind regards,

on behalf of

Dr. Sonu Bhaskar

Academic Editor

PLOS ONE